# Overcoming Current and Preventing Future Nosocomial Outbreaks during the COVID-19 Pandemic: Lessons Learned at Three Hospitals in Japan

**DOI:** 10.3390/ijerph181910226

**Published:** 2021-09-28

**Authors:** Makiko Komasawa, Myo Nyein Aung, Kiyoko Saito, Mitsuo Isono, Go Tanaka, Saeda Makimoto

**Affiliations:** 1Ogata Sadako Research Institute for Peace and Development, Japan International Cooperation Agency, Shinjuku-ku, Tokyo 1628433, Japan; Saito.Kiyoko@jica.go.jp (K.S.); Isono.Mitsuo@jica.go.jp (M.I.); Makimoto.Saeda@jica.go.jp (S.M.); 2Advanced Research Institute for Health Sciences and Faculty of International Liberal Arts, Juntendo University, Bunkyo-ku, Tokyo 1138421, Japan; myo@juntendo.ac.jp; 3Human Development Department, Japan International Cooperation Agency, Chiyuda-ku, Tokyo 1028012, Japan; Tanaka.Go@jica.go.jp

**Keywords:** SARS-CoV-2, COVID-19, hospital, Japan, nosocomial infection, outbreak, asymptomatic transmission, infection prevention control, health system

## Abstract

Hospitals are increasingly challenged by nosocomial infection (NI) outbreaks during the ongoing coronavirus disease 2019 (COVID-19) pandemic. Although standardized guidelines and manuals regarding infection prevention and control (IPC) measures are available worldwide, case-studies conducted at specified hospitals that are required to cope with real settings are limited. In this study, we analyzed three hospitals in Japan where large-scale NI outbreaks occurred for hints on how to prevent NI outbreaks. We reviewed openly available information from each hospital and analyzed it applying a three domain framework: operation management; identification of infection status; and infection control measures. We learned that despite having authorized infection control teams and using existing standardized IPC measures, SARS-CoV-2 may still enter hospitals. Early detection of suspected cases and confirmation by PCR test, carefully dealing with staff-to-staff transmission were the most essential factors to prevent NI outbreaks. It was also suggested that ordinary training on IPC for staff does not always provide enough practical knowledge and skills; in such cases external technical and operational supports are crucial. It is expected that our results will provide insights into preventing NI outbreaks of COVID-19, and contribute to mitigate the damage to health care delivery systems in various countries.

## 1. Introduction

The first confirmed case of the coronavirus disease 2019 (COVID-19) in Japan was in Tokyo on 24 January 2020 [1]. The first wave started in April, followed by the second wave in September, and the third wave in December 2020 (Figure 1). The number of infected people increased over the period leading up to the third wave with the total of hospitalized patients peaking at 12,128 in Japan as of 16 March 2021, of which 377 were in critical condition [1]. As the number of community-acquired infections (CAIs) of unknown route increased, risk of a nosocomial infection (NI) outbreak increased [2,3].

In Japan, cases where five or more people are infected in a single chain of close contacts are defined as a “cluster” [4]. Although there are no official statistics on NI clusters in Japan, Oshitani, H. counted that 321 clusters occurred in medical and long-term care facilities from December 2020 to January 2021, and the number of infected people reached 8,191 [4]. The Japanese Association for Infectious Diseases also reported that, as of December 2020, out of 263 facilities in Japan, 16% of health or welfare facilities had experienced NIs and 58% of them expanded into clusters [5]. More than half (52%) contained the NIs within a week while 5% of them remained infected for more than four weeks. Although a great deal of experience in infection control for SARS-CoV-2 has been accumulated and shared, large-scale clusters continue to occur [6,7,8,9,10,11,12,13]. The tension around infection control (IC) in hospitals is increasing. Once a cluster occurs, hospitals need to take measures, such as suspending outpatient functions and new hospital admissions. This results in a significant decline in hospital functions and, in turn, disrupts healthcare delivery in the region.

Although various guidelines and manuals have been globally published regarding infection prevention and control (IPC) measures for COVID-19 in healthcare facilities [14,15,16,17,18,19], it seems that learning through standardized documents or desk-top trainings may not be practical to deal with real emergency settings. 

We analyzed large-scale NI outbreaks of COVID-19 that occurred in Japanese hospitals and examined what factors caused the outbreaks, and what countermeasures were effective in order to stop the chain of infection in the hospitals.

## 2. Materials and Methods

### 2.1. Study Design

This is a comparative study using openly published information concerning three hospitals that experienced considerable NI outbreaks in the first year of the COVID-19 pandemic in Japan. All of the information we used is available in “Appendix A in Appendix A. The observation period began from the day (Day 0) that the first symptomatic case confirmed by PCR testing, “index case,” to the day that the hospitals declared containment. We reviewed the process of outbreaks, appraised the surveillance data, and analyzed descriptive data in the hospitals. Then, we compared key factors of NI control measures across the three hospitals. 

### 2.2. Study Target

The three hospitals were regional-based, middle-sized, acute care hospitals without any beds designated for infectious diseases (Table 1). None of them had a laboratory for a polymerase chain reaction (PCR) test at the time of the outbreak. Hospital A was assessed as the first large-scale NI outbreak in Japan [20]. Hospital B experienced a large-scale NI in its local city where simultaneous outbreaks of large-scale clusters occurred in three core hospitals [9,21]. Hospital C encountered the largest NI outbreak in Japan as of 1 March 2021 [10]. The timings of each outbreak are shown in Figure 1.

Hospital A is a core hospital in Taito Ward (population: 203,827, as of 1 August 2021) [22], 99 part-time doctors have been dispatched from the university hospital as of 21 April 2020 [23]. Hospital B is a general hospital in Asahikawa City (population: 329,033, as of 1 August 2021) [24] in Hokkaido, and 35% of its beds are occupied by elderly patients with a high level of care. Hospital C is a general hospital in Toda City (population: 141,154, as of 1 August 2021) [25] in Saitama Prefecture, one of the bed towns within metropolitan Tokyo.

### 2.3. Setting

Infectious disease control in Japan is regulated by the Infectious Diseases Control Law. Legislation to classify COVID-19 as a designated infectious disease was implemented on 1 February 2020 (Cabinet Order No. 11 of 2020) [26]. The public health centers (PHC) are in charge of its control. It oversees surveillance, contact tracing, instructions for receiving PCR tests and diagnosis, coordination of hospitalization and isolation, and so on. This is a different approach from the normal medication under the medical insurance scheme.

Since 1996 all hospitals in Japan prohibited family and other outside member care; all care is given by in-house professional staff [27]. As a result, access by family members and visitors to the inpatient wards has been restricted; whereas, the outsourcing of non-medical services in hospitals has been progressing (e.g., cleaning, linen laundry, hospital meals, cafeteria, etc.). These contracted staff are allowed to enter the wards.

All three hospitals were authorized as hospitals prepared for infection prevention and control (Kansen-boushi-taisaku-kasan) by the Ministry of Health, Labour and Welfare (MOHLW) [28]. This indicates they have established an infection prevention and control department with an infection control team (ICT). The team includes a full-time physician with at least three years of experience in infection control, a full-time nurse with at least five years of experience in infection control who has completed training in infection control, a full-time pharmacist with at least three years of hospital experience, and a full-time laboratory technician with at least three years of hospital experience. The hospitals each have an individual manual for IPC, and provide training on IPCs for the staff on a regular basis.

### 2.4. Analytical Framework

We developed the analytical framework for this study based on the “Checklist for Countermeasures to Combat Infections of Novel Coronaviruses in Hospitals and Facilities” published by the Japanese Society for Environmental Infection Research (22 July 2020) [29]. The framework includes three domains: operational management; identification of infection status; and infection control measures. Detailed check points are shown in Table 2.

## 3. Results

### 3.1. Hospital A

#### 3.1.1. Outbreak timeline at Hospital A

The duration of the outbreak in Hospital A was from 23 March (Day 0) to 8 June (Day 77) in 2020 [20,30]. The total number of infected cases was 214, consisting of 131 inpatients and 83 staff, with 43 inpatient deaths as of 1 July 2020. Table 3 shows the timeline of events related to the outbreak in Hospital A.

#### 3.1.2. Operational Management at Hospital A

Hospital A promptly contacted the PHC as soon as suspicion of a NI outbreak became apparent, and took countermeasures while seeking instructions. In addition, since the first large-scale outbreak in Japan was expected, the MOHLW cluster team was dispatched to the site from Day 7 to conduct an active epidemiological survey. The cluster team identified the route of infection and published an interim investigation report on the results on Day 23. In addition, the team provided guidance on countermeasures against the spread of infection tailored to the site [20,30].

The hospital ICT had been vigorously working IPC activities prior to the NI pandemic occurred [20]. During the outbreak, they led the daily infection monitoring and staff reassignment planning. In the beginning, decisions made by upper management were not immediately conveyed to the frontline staff, which increased their sense of anxiety. Therefore, the detailed information/instructions were written each time on a large board on the walls of the corridors where staff could see it to ensure prompt communication with all frontline workers [31].

In order to prevent the spread of infection, Hospital A suspended outpatient services and new admissions in wards with COVID-19 suspected patients from Day –2. From Day 2 the hospital stopped accepting new admissions for the entire hospital.

Various supports for staff were also implemented. In April, when the NI began spreading, a mental health support team was set up. The team consisted of a psychiatrist, a neurologist, a clinical psychologist, and a pediatrician [31]. The team conducted regular surveys and offered counseling for psychological conditions, and sent the patients to consultations if necessary. The hospital also provided wide-ranging support for its staff, including special days off for employees who could not work due to the quarantine, salary guarantees for those who were absent from work, and accommodation for those who could not return home. In addition, the hospital protected staff from excessive fact finding by journalists [31]. Furthermore, the hospital’s doctors voluntarily set up a crowdfunding campaign and the accommodation expenses of the health care workers (HCWs) were subsidized [20,31].

#### 3.1.3. Identification of Infection Status at Hospital A

Figure 2 shows the epi curve of 130 cases (82 inpatients and 48 staff) with confirmed onset dates and estimated onset dates by the MOHLW cluster team [30]. It illustrates that many infections had already occurred before Day 0 and indicates a trend of infection occurring first in patients and then spreading to staff [30].

The epidemiological study by the MOHLW cluster team estimated that the original infected case appeared nine days before the index case [30]. One possibility for the original case was a patient with cerebral infarction admitted on 26 February. As the patient had repeated fever since 5 March [8], it was assumed that the NIs had already spread in the ward by Day 0. The main reason for the delay in detecting NIs was the lack of awareness among the HCWs about the characteristics of COVID-19. Namely, that COVID-19 is already highly contagious before the onset of symptoms, and that even if infected, the patient is often asymptomatic [20,31]. In addition, in acute care hospitals, it is common for patients to develop fever and pneumonia due to the primary disease or its treatment. This makes it difficult to distinguish the symptoms from those of COVID-19. The PCR test is essential for identifying COVID-19 infection in such cases. However, the hospital did not have a laboratory for PCR testing and the capacity for PCR testing in Tokyo was low at that time; it took nine days to obtain all the results [20]. The hospital deduced that the outbreak within the hospital spread during the time waiting for the PCR results [20].

The first outbreak occurred in Ward A of the west wing on the fifth floor and then spread to Ward B of the east wing. The two wards are located on the same floor, adjacent to each other via an elevator hall (Figure 3). Before Day 5, HCWs and patients in these wards could move across the wards without being confirmed negative by PCR test, which may have caused the spread of the infection [30]. Initially, it was expected that the outbreak would be contained on the fifth floor [32], but it later expanded to Ward G (hematology) on the eighth floor, which became the center of the outbreak. This was presumed to have originated from patients transferred from Wards A or B [30]. In addition, as the number of infected HCWs increased, more and more staff members were in charge of multiple wards, and it was speculated that there was cross-infection among wards via staff. Furthermore, the average age of infected HCWs was as young as 35 years (as of Day 47), and many of them were asymptomatic, suggesting that HCWs may have unknowingly become infection vectors [30]. In addition, the possibility of transmission via in-hospital movement of patients with dementia or those who have difficulty wearing a mask at all times and whose behavior is difficult to control was also pointed out [30]. Of the 43 patients who died, 23 were hematology patients. It is believed that lowered resistance due to the primary disease, its treatment, or a delay in response led to the seriousness of the disease [20,31].

#### 3.1.4. Infection Control Measures at Hospital A

After it became apparent there was an outbreak, Hospital A took measures such as suspending outpatient services and new hospital admissions, and isolating HCWs in the ward where the outbreak occurred [30].

Before Day 5, several contamination risks were observed while staff were wearing PPE. First, many HCWs tended to keep wearing PPE after completing their work in the designated areas, which increased the risk of spreading the infection to the surrounding environment. Second, working in PPE increased HCW’s fatigue and stress, and led to incomplete performance of basic infection control activities, such as hand hygiene [33]. Therefore, from Day 5, the strategy was changed [33]. All medical and non-medical staff working on floors 5–8 were required to wear PPE. They put on PPE on floor 3 and then moved to the designated floor by elevator. They were then required to remove PPE before returning to floor 3. The PPE was comprised of waterproof gowns, waterproof pants, shoe covers, caps, gloves (double), N95 respirator masks, surgical masks, and face shields [33]. In addition, a long-sleeved gown was worn during direct patient care, and the outer gloves and gown were changed for each patient [33].

Up until Day 8, PCR-positive and PCR-negative patients were mixed in several wards, even though they were separated by room. Due to the possible risk for transmission, the hospital decided to cohort the two types of patients, the patients with/without a confirmed COVID-19, by ward completely [33]. Symptomatic-negative patients were also isolated (pool-period) and frequent PCR testing was conducted. They were transferred to red zone beds if the results turned positive. Furthermore, on the fifth to eighth floors, only the inpatient rooms were considered as contaminated areas and other areas were classed as clean areas [33]. This was to minimize the contaminated areas and mitigate the HCWs’ stress. These reorganizations took 10 days [8,30]. The rules for wearing PPE were also changed. In all wards, it was made a rule to wear PPE when entering inpatient rooms and to take it off when leaving. Since there tended to be a shortage of PPE, N95 masks were reused as needed while taking measures to prevent contamination [33].

After the staff-to-staff transmissions were pointed out, preventive measures were expanded [20,31]. All staff members were required to take their temperature before going to work and to take time off if they had a fever. In addition, in order to reduce the risk of cross-infection among staff members, the following measures were taken: avoid close contact among staff members in staff rest rooms, napping rooms, locker rooms, etc. (specifically, wearing masks during breaks and eating and drinking, and prohibiting face-to-face eating) [20]. Thorough education on IPC measures was provided [20]. Not only doctors and nurses, but also co-medical and administrative staff involved in ward operations were targeted for training [33]. The content of the training included the timing and method of hand disinfection, appropriate PPE selection and removal, and disposal of medical waste [33].

Measures to thoroughly clean and disinfect the hospital environment were also implemented [33]. Frequent wiping was performed, especially on high frequency contact surfaces. Electric doors were installed between the West Wing and the East Wing (Figure 3), where staff and patients come and go on the same floor [20].

### 3.2. Hospital B

#### 3.2.1. Timeline of Outbreaks at Hospital B

Hospital B’s outbreak duration was from 6 November 2020 (Day 0) to 30 January 2021 (Day 85) [9]. The total number of infected cases was 214, including 136 inpatients and 78 staff, with 36 inpatients death as of 20 January 2021. Table 4 shows the timeline of events related to the outbreak in Hospital B. All information regarding Hospital B was extracted from reference [9] unless otherwise specified.

#### 3.2.2. Operational Management at Hospital B

Hospital B contacted the PHC as soon as the first PCR-positive case was confirmed, and in the afternoon of Day 0, a taskforce team was set up in the hospital by both Asahikawa and Kamikawa PHCs (4 members). From Day 1, a Disaster Medical Assistance Team (DMAT) from Japanese Red Cross (JRC) hospital in Asahikawa (2 members), a specialist in cluster analysis from Hokkaido Provincial Government (1 member), and hospital staff (five members) joined the task force. After this, NI control measures were implemented with the taskforce team under the direction of the PHC. Hospital B stopped its outpatient and health checkup services from Day 3.

The first external expert team was dispatched on Day 14. It included members from the JRC-DMAT, the Health Services University of Hokkaido (HSU), and the National Institute of Infectious Diseases (NIID). The second team was comprised of epidemiologists from DMAT-Osaka, the Self-Defense Forces, HSU, and NIID. It was dispatched on Day 21. They analyzed the current infection status and evaluated countermeasures, decided on IPC strategy, and implemented thorough infection prevention measures in the hospital. The HSU professor also provided hands-on training related to PPE. Nurses from the Self-Defense Forces and other organizations were also dispatched for 2–3 weeks, based on the hospital’s request.

Since Hospital B is not a designated hospital for infectious diseases, if a patient gets infected, the patient needs to be transferred to a designated hospital. In the case of Hospital B, the transfer of PCR-positive patients to the designated hospital started from Day 1. However, as CAIs spread in the city around Day 13, it became difficult to transfer newly infected patients. In response to this, DMAT directly requested transfers to the five core hospitals in Asahikawa. Moreover, a meeting of the directors of the five core hospitals in Asahikawa was held, and it was agreed that ten patients would be transferred to the hospital.

As the number of infected staff increased rapidly, the hospital requested the PHC to isolate them in a designated accommodation, but the request was denied. Therefore, Hospital B converted 11 beds in an intensive health check-up room to be used as an isolation facility for the staff. In response to a decrease in the number of HCWs to one-third since mid-November, the dispatch of external nurses such as the Self-Defense Forces was requested.

When the spread of the infection was extremely severe on Day 19, the hospital representative met with the Mayor of Asahikawa City and requested assistance, including (1) dispatch of 20 or more nurses, (2) disinfection and cleaning, (3) disposal of medical waste, and (4) supply of PPE, but this request was not accepted [9]. On the same day, they sent a similar request to the Governor of Hokkaido.

#### 3.2.3. Identification of Infection Status at Hospital B

At Hospital B, one patient and one staff member on the sixth floor’s ward were confirmed to be PCR positive on Day 0. In response to this, PCR testing of all close contacts was conducted until late at night on the same day according to the instructions of the PHC. As a result, 21 patients and 4 nurses on the sixth floor were found to be positive, and several doctors in charge were found to be negative. This was the beginning of a large-scale NI outbreak. Figure 4 shows the epi curve of cases by PCR confirmation date in Hospital B up until the end of the available data. Initially, the infected were mainly patients, but later the number of infected staff increased. On the other hand, the number of newly infected staff peaked on Day 32 and newly infected patients peaked on Day 41.

In the sixth floor ward, the number of patients with fever became noticeable around one week before Day 0 (one patient on Day –6, two patients on Day –4, one patient on Day –2, six patients on Day–1, and six patients on Day 0). Later, all of these febrile patients were found to be PCR positive. It was noted that the timing of suspecting COVID-19 infection was too late. Many of the patients who developed the disease were elderly patients who required full assistance with dressing, eating, and excretion, which inevitably required intensive close contact care, and it was assumed that the tiny error in infection prevention (time of changing PPE for each patient or disinfecting nursing-care equipment) could have led to the spread of the infection. In fact, the breakdown of the new positive cases during the first ten days shows that out of 43 cases, 39 were elderly patients with full care needs. 

Initially, the PHC expected the outbreak to end on the sixth floor ward, but in fact it spread immediately to the seventh and fifth floors and later to the fourth floor, the first floor, the second floor, and finally to all floors by Day 18 (Figure 5). Cross-infection in patients-to-staff and staff-to-staff within the wards or across the wards was analyzed as the cause.

In mid-November, the PHC instructed the hospital to switch from PCR testing to antigen testing for suspected infected patients, due to the fact that two other hospitals in Asahikawa City were also experiencing outbreaks and the PCR testing work was becoming burdensome in the city.

Three weeks after the last confirmation of infected case on Day 58, the NI infection was declared contained. There were 78 cases of infection among the staff. Seventeen cases involved non-medical staff; none of whom had any contact with the ward or with ward duties. All staff members followed basic precautionary measures and had adequate ventilation in the rooms. Detailed close contact-tracing and genomic analysis of the virus were necessary to determine the infection routes, but these had not been done, and the route of entry of the NI was not detected as of Day 79 [21]. The number of deaths from COVID-19 was 39 patients (no detailed breakdown of information was available).

#### 3.2.4. Infection Control Measures at Hospital B

Initially, it was expected that the NIs would be contained only on the sixth floor and all measures, including PCR testing and zoning, wearing of masks for droplet control, hand disinfection, and precautions against cross-infection among staff were concentrated on the sixth floor. However, on Day 5, a patient on the seventh floor was confirmed to be positive, and on Day 9, a patient on the fifth floor was confirmed to be positive, and the spread of the infection between wards became noticeable. Therefore, from Day 9, all HCWs were required to wear PPE (i.e., apron, mask, Tyvek, gloves, face guard, cap, gown) on the fifth to seventh floors. Additionally, because of the risk of cross-infection among staff dispatched to several wards, the staff were assigned to a fixed ward. HCWs and non-medical staff were strictly zoned (e.g., changing rooms, rest rooms, and inside the wards). 

In order to reduce the risk of infection of doctors, the ward doctors limited their rounds, and only nurses provided care in the ward, by the instructions of the taskforce team from Day 9. However, during Days 16 to 19, five chief nurses in wards became PCR-positive, one after another. Therefore, on Day 24, after intensive PPE wearing instruction by a professor of HSU, doctors resumed their ward rounds. Consequently, the doctors were able to monitor the patients’ conditions closely and provide prompt treatment.

During Days 14 to 18, an external expert team was dispatched again to evaluate the IPC measures. Practical on-the-job training on infection control, ward and hospital cleaning and disinfection (including ultraviolet (UV) irradiation) for individual staff members were conducted. At this point, all floors housed both infected and non-infected patients together [34]. The hospital gradually made the fourth and seventh floors into the non-contaminated zones, by transferring some positive patients to other hospitals, and enabling a triage within the hospital. The outsourced cleaning service was terminated, and non-medical staff began cleaning, disinfecting, and managing the trash. The expert team set up a comprehensive chain of command with all wards. Meetings were held twice a day, morning and evening, with representatives of all wards, to quickly grasp and share the current status of hospital infection, patient transfer, human resources, supplies, garbage, linen, food, and staff care.

The cohorting on the sixth floor was tightened by the Self-Defense Force nurses and the external support nurses dispatched on Day 32, and the roles of staff were clarified. Individual IPC practices (PPE removal methods, hand hygiene, etc.) were frequently supervised by the external nurses. Around this time the IPC measures became pertinent to each zone. Furthermore, detailed symptoms of all patients (oxygen saturation, chest CT-scan, etc.) were documented by room and this information was utilized in the management of patients and beds. The hospital’s disinfection status was improved by frequent UV irradiation, cleaning, and sterilization. Around Day 41, NI control measures stabilized.

### 3.3. Hospital C

#### 3.3.1. Timeline of Outbreaks at Hospital C

The period of Hospital C’s outbreak was from 27 November 2020 (Day 0) to 26 February 2021 (Day 91) [10]. The total number of PCR-positive cases was 318, including 149 patients and 169 staff, with 31 patients dead as of 20 January 2021. The infection spread to 12 wards out of 16 wards [11]. Table 5 shows the timeline of events related to the outbreak at Hospital C.

#### 3.3.2. Operational Management at Hospital C

Hospital C initially thought that it could control the outbreak by itself [11]. However, on Day 38, it determined that it would be difficult to control the outbreak on its own, and requested assistance from Saitama Prefecture. As a result, a Joint Taskforce Team (JTT) was established by the Saitama Prefectural Government, comprising the Nanbu PHC, the cluster control team of MOHLW COVID-19 Taskforce and the regional support team of MOHLW COVID-19 Taskforce [10]. The JTT spent three weeks trying to identify the route and cause of transmission of the infection, and made recommendations on how to strengthen IPC measures and how to resume hospital services [35]. 

Hospital C stopped admitting new patients to the infected wards on Day 0 and stopped admitting new inpatients and new outpatients in all wards on Day 25. However, outpatient re-examination services were continued. In accordance with the JTT strategy, a collaborative system with Tokyo Medical University was established. In addition, support staff (20 people) were dispatched from the partner hospitals [10].

#### 3.3.3. Identification of Infection Status at Hospital C

Figure 6 shows the epi curve of confirmed cases by PCR confirmation date in Hospital C. It showed that the infection was initially sporadic among the staff, but from Day 21, the infection spread to inpatients, indicating that the hospital had entered a phase of the outbreak.

According to the results of the JTT investigation, one of the factors reported was that there were several patients whose PCR results were initially negative but later had positive PCR results due to the longer incubation period of the COVID-19 [35]. In addition, there had been a rapid increase in the number of CAI of COVID-19 in Saitama Prefecture, suggesting that it may have been brought in by patients or staff. The presence of inpatients who had difficulty wearing masks at all times has also been reported as a factor. However, even with the JTT investigation, the details of the source and route of infection have not been clarified (as of 21 February 2021) [12].

On the other hand, as for the staff, the number of infected staff reached 174, which is nearly 10% of all staff [11]. It was pointed out that although standard precautionary measures (masks, hand disinfection, temperature check) had been taken from the beginning, risk of transmission were observed. For example, conversations without masks in break rooms or cafeteria, close contact when assisting in removing PPE, fatigue from wearing heavy PPE for long periods of time, or close-contact care with patients were reported [35].

An increase in the number of infected patients in Saitama Prefecture and a shortage of beds for infectious diseases in the prefecture meant most of the PCR-positive patients could not find a place to be transferred to and had to be treated continuously in their own hospitals. Such a situation might lead to inappropriate zoning and increased risk of infection in the hospital [35]. After the JTT’s intervention, the number of new infections rapidly decreased (0–4 new cases per day), and after the last case occurred on Day 64, no new NI were confirmed for four weeks [11].

#### 3.3.4. Infection Control Measures at Hospital C

Four measures were taken in response to the JTT verification results: (1) strengthen early detection and early response; (2) reorganize zoning and strengthen IPC; (3) strengthen HCW management, and (4) collaborate with infectious disease experts from Tokyo Medical University [10]. Specific measures for point 2 include strictly designated PPE removal areas according to zoning, providing thorough guidance on use of PPE, strengthening hand hygiene, taking measures to prevent the risk of infection among staff (e.g., staggering break times, prohibiting conversation in staff rooms, keeping distance in cafeteria), and providing PPE to laboratory technicians and rehabilitation staff who come into direct contact with patients on a regular basis. Concerning point 3, staff from partner hospitals were dispatched and guidance for staff returning to work after isolation were implemented.

The summary of remarkable risk factors in the spread of NI and factors in its prevention in the three hospitals is shown in Table 6.

## 4. Discussion

This study revealed that despite having authorized ICTs and using standard IPC measures, NI outbreaks may still occur in a hospital. Our findings illustrated essential solutions to address major pitfalls including: (1) early detection of suspected COVID-19 infection cases, immediate confirmation by PCR testing to identify infection status; (2) implementation of effective cohorting and zoning; (3) raising awareness of the risk of staff-to-staff transmission in relation to infection controls are key to prevent NI outbreaks. The results also underscored the importance of strengthening coordination and guidance by the PHC and local governments as well as other hospitals, and assistance from external experts, especially for the hospitals that have no experience responding to NIs of COVID-19.

### 4.1. Early Detection of Suspected Cases

At the three hospitals, one of the most salient problems being highlighted was the delay in suspecting infection with the SARS-CoV-2 and the delay in conducting PCR testing. A remarkable challenge of COVID-19 is that many infected people are asymptomatic [2,15,36,37,38]. For example, in the case of the COVID-19 outbreak on the cruise ship Diamond Princess, 58% were asymptomatic [39]. The other Japanese study found that 96% of PCR-positive cases were transmitted from asymptomatic sources [40]. It has also been noted that the incubation period is longer and the infection is highly contagious before the onset of symptoms [41,42]. Early recognition is the most important countermeasure [14]. In both Hospitals A and B, because elderly patients with a fever caused by the primary disease had symptoms similar to COVID-19, the infection was overlooked for several weeks. Thus, it was assumed that the infection had spread in the hospital during that time. In addition, at Hospital A, a later intensive epidemiological examination revealed that the index case had been brought in by a patient at the time of new admission. It is recommended that PCR testing be performed on admission and that isolation be maintained until test results are available [17,42]. Because of the false-negative period, it is recommended to repeat the PCR testing [42]. One thing we could emphasize is that PCR tests should be available when necessary [42]. The case of Hospital A, where the test was not available or it took a long time to get the results must be avoided. However, if PCR testing is not available, using a combination of in-hospital antigen testing, imaging and physician diagnosis, as was done in Hospitals A and B, may be considered [17].

### 4.2. Transmission among HCWs

Routes of infection can be patient-to-staff, patient-to-patient, and staff-to-staff [43]. The previous study in Japan reported that the staff-to-staff route accounted for 31% of NIs [40]. The three cases in this study revealed that the risk of cross-infection among staff was not sufficiently recognized at the beginning of the outbreaks, this was one of the factors contributing to the spread of the infection. In particular, the average age of HCWs is younger than that of the general population. The younger the HCWs, the higher the rate of asymptomatic infections, and thus the higher the probability that they are unknowingly carrying the virus [42]. In the case of Hospitals A and B, it was noted that increased fatigue from heavy pressure to perform strict IPC allowed the virus to enter the wards. It was also observed that the infection was acquired in nonclinical settings (i.e., by crossing the lines of traffic between staff, and by conversations in changing rooms and cafeterias without wearing masks or not keeping enough distance) [2,40,44]. In Hospital C, it was pointed out that staff may have brought the virus in from outside. In fact, several studies suggested that HCWs can be infected in the community and possibly amplify SARS-CoV-2 outbreaks in the health facilities [3,45]. The Osaka study implied that the risk of bringing the virus in from outside by staff tends to increase as CAI via unknown routes increase, and it takes around 10–20 days until SARS-CoV-2 can be recognized [3]. Therefore, they suggested that, by paying attention to the number of unknown route cases in the community, it may provide a welcome trigger to start stricter prevention inside the hospital. In the case of the top referral hospital in Vietnam, PCR testing for early detection of CAIs was conducted not only on all staff but also community residents [2]. This initiative required strong political and administrative leadership [2]. In addition, the major source of NIs in the Vietnamese hospital was outsourced-workers in the cafeteria [2]. It recommended that management should be acquainted with the working schedules and movement within the hospital of all staff, including the outsourced staff from external contractors [2,3,5].

### 4.3. Care for HCWs

HCWs tend to get infected, especially front-line nurses who are regularly in close contact with infected patients whilst administering physical care [43,46,47]. According to a previous survey in Japan, 52% of the staff infected were nurses [40]. In the three hospitals in our study, when an infection is found in a ward, the nurses in close contact with the relevant people are isolated immediately, which means that ward duties must be performed on a limited shift, increasing the workload per person [18,30,43,45,46,48,49]. In addition, when shift changes required support from other departments, the work was unfamiliar as were the new members, which increased stress [31,49]. The burden on the remaining nurses should be reduced as much as possible by actively recruiting nurses from outside the organization whenever possible. 

In Japan, in particular, 92% of nurses were women in 2018 [50]. Women are more likely burdened with varied non-professional and family-related burdens, such as anxiety about the spread of infection to family members, sense of guilt about restrictions on family life, and cases where they are forced to leave their jobs due to living with elderly family members, family members with underlying medical conditions, or the fear of stigma [31,43,51]. Many nurses managed to overcome the situation by their sense of social mission, but there were more than a few nurses on the verge of burnout [31]. Infectious disease outbreaks are similar to disasters: serious symptoms such as psychological trauma often appear after the outbreak is over so that ongoing care is necessary [31]. Further studies are required on the mental health and wellbeing of HCWs during and after a pandemic [51,52].

### 4.4. Support from Outside Experts

The three hospitals each have an ICT, an individual manual for their own facility, and appropriate training for their staff on a regular basis. However, in the initial response, the discovery of the infection was delayed, the spread of the infection was poorly predicted, and the management of cohorting and zoning were not appropriate. In Japan, the PHCs are responsible for infection control in both community and facilities. Hospital C’s delay in requesting assistance from the PHC was the most serious error in its strategic decision. Responding to the NI outbreak requires various types of technical expertise. This includes epidemiological analysis, zoning and cohorting, precise PPE management, and environmental disinfection. Our findings highlighted that immediate practical guidance and intensified training on-site by experts with experience were effective to make all staff familiar with strict IPC measures in hospitals less experienced in NIs, as the Hong Kong study stressed [45]. Moreover, its study suggested that the on-site simulation drill in normal times was useful [45]. In Japan, at present, the cluster teams of MOHLW and the infectious disease specialists of DMATs affiliated with each prefecture play an important role in COVID-19 control and training HCWs to deal with on-site outbreaks. Considering the need to prepare for unknown infectious diseases in the future, strengthening human resources to include highly experienced specialists is imperative.

### 4.5. Operational Management

Our findings implied that enhancing operational management were essential for controlling NI in emergencies, although not many guidelines mentioned this aspect. First, establishing a taskforce team with the PHC authorities and external experts resulted in robust progress toward NI control in Hospital A and B; however, Hospital C failed to get immediate external support which prolonged the period of the outbreak. Second, establishing an operational command system and directly communicating with the staff in Hospital A and B was effective during the NI outbreaks. Hospital B set up a mechanism for issuing instructions and up-to-date information exchange was established by gathering representatives from all departments. Hospital A posted information on the wall of a busy corridor in order to communicate changing instructions to the staff as quickly as possible. Third, securing materials and human resources (especially nurses), as well as places to transfer PCR-positive patients and isolated accommodation for infected staff was also an important task for management. In Hospitals B and C, securing a place to transfer PCR-positive patients was a big challenge when the CAIs spread. Hospital B struggled to provide a safe and stress-free place for quarantine for the infected staff. For transferring infectious patients and securing quarantine facilities for staff, a close relationship with other hospitals and the local authorities in the region is crucial; and developing better relationships with them at normal times is also imperative [2,18,45]. Further studies on coordination and reallocation of roles of each hospital in the region for tackling with NI outbreaks should be undertaken. 

### 4.6. Limitation

There are several limitations to this study. First, the analysis was based only on secondary information, which limited the accuracy and scope of the information. For example, we did not have enough information to draw a complete epidemic curve for each hospital. Second, the analysis was limited to three cases, and thus cannot be generalized. Third, this study focused on the analysis from a public health perspective and hardly touched on clinical aspects. Furthermore, we could not adequately discuss the following important perspectives: (1) the scope and frequency of PCR test for screening during the outbreaks; (2) timing and scale of hospital service disruptions and criteria for resumption during the NI outbreak; (3) the physical structure of each facility (e.g., air conditioning, staff movement lines, etc.); (4) criteria for containment of NI outbreak; and (5) how to appropriately deal with the spread of CAIs and the occurrence of NI outbreaks: thus further studies are required. Finally, we focused on the clusters in the first year of the COVID-19 pandemic and not on the second year in Japan, which may limit our insight. Despite the above limitations, our study provided hints for preventing SARS-CoV-2 from entering hospitals and minimizing the spread of NIs.

## 5. Conclusions

Our points of recommendation highlighted some of the essential factors surrounding the control of NI outbreaks. It is almost impossible to keep SARS-CoV-2 out of a hospital. The key strategies to tackle NI outbreak of COVID-19 are as follows: to detect the virus as soon as it enters the hospital, prevent the spread of infection in the facility, to strictly practice basic IPC measures with the intensive support of outside knowledgeable and experienced experts under the coordination of the PHC. The lessons learned from the Japanese hospitals may provide useful insights for healthcare facilities and regional health administrators in both developing and developed countries. It may help them to mitigate the damage to health delivery systems caused by NI outbreaks of COVID-19. 

## Figures and Tables

**Figure 1 ijerph-18-10226-f001:**
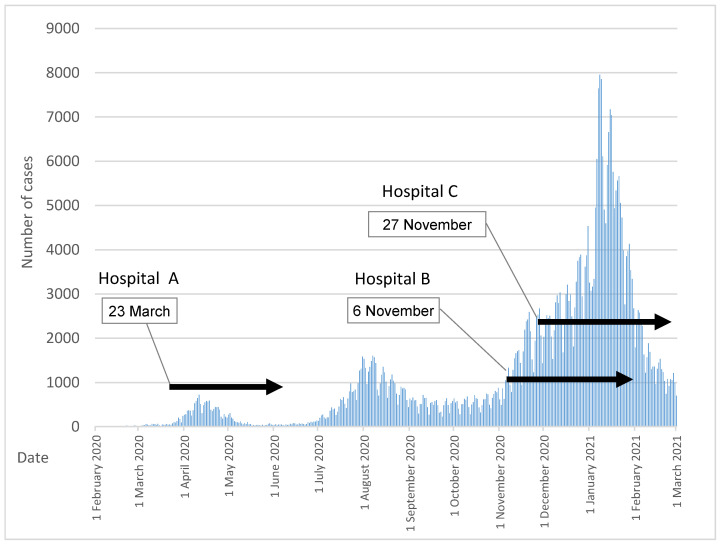
Spread of COVID-19 infection and the timing of the three hospital’s outbreaks. Authors made based on https://www.mhlw.go.jp/stf/covid-19/kokunainohasseijoukyou.html#h2_1, accessed on 1 April 2021.

**Figure 2 ijerph-18-10226-f002:**
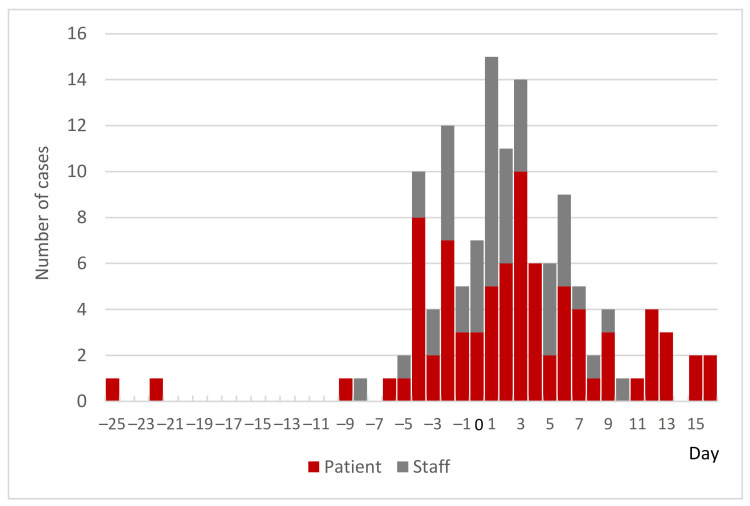
Epi curve of cases with confirmed onset dates and estimated onset dates at Hospital A. Cases confirmed between Day −25 and Day 41 (*n* = 130).

**Figure 3 ijerph-18-10226-f003:**
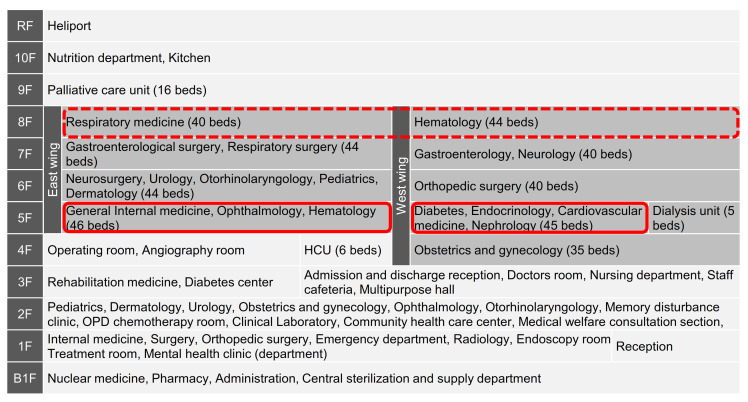
Ward map and major sources of COVID-19 infection of Hospital A. Shaded parts indicate the wards that became infected zone. The west wing of the fifth floor was the ward in which the first case was found and spread to the east wing. The dotted line on the eighth floor indicates the wards that became the second center of infection. Authors made based on the hospital website, http://www.eijuhp.com/floor_guide.html, accessed on 1 April 2021. HCUs, high care unit; OPD, outpatient department.

**Figure 4 ijerph-18-10226-f004:**
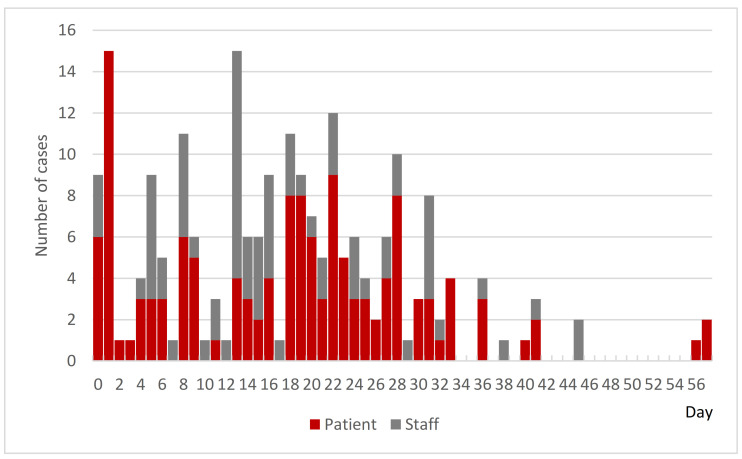
Epi curve of cases by PCR confirmation date at Hospital B. Cases confirmed as PCR positive between Day 0 and Day 57 (*n* = 213).

**Figure 5 ijerph-18-10226-f005:**
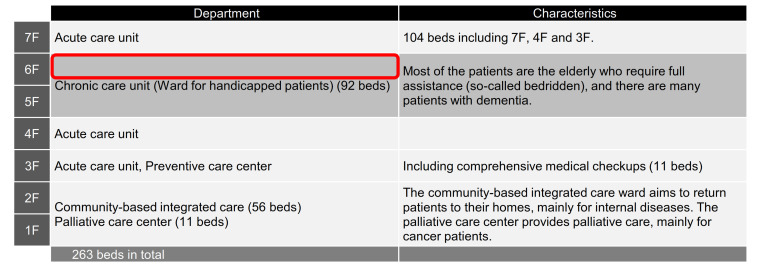
Floor map of Hospital B with characteristics. The shaded part indicates the ward with a chronic care unit. The bold line on the sixth floor indicates the ward with the index case.

**Figure 6 ijerph-18-10226-f006:**
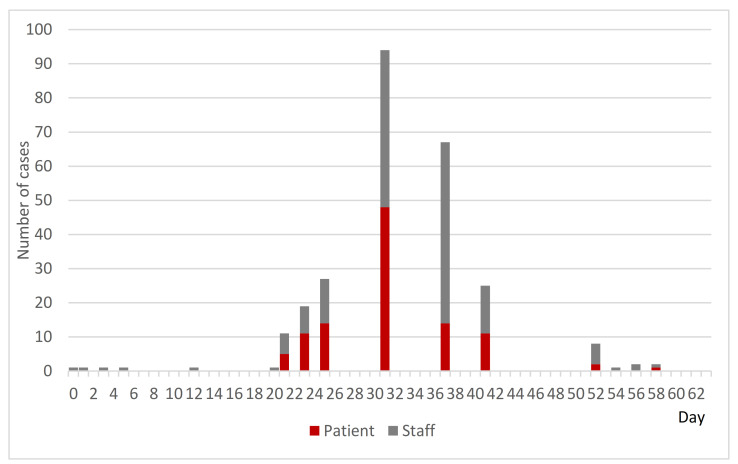
Epi curve of cases by PCR confirmation date at Hospital C. Cases confirmed as positive by PCR between Day 0 and Day 58 (*n* = 262). Due to limited available data, some bars showed cumulative numbers of several days instead of daily numbers.

**Table 1 ijerph-18-10226-t001:** Characteristics of three hospitals.

	Hospital A (As of 8 June 2020)	Hospital B (As of 20 January 2021)	Hospital C (As of 20 June 2021)
Location	Taito Ward, Tokyo	Asahikawa City, Hokkaido	Toda City, Saitama
Operating entity	Private	Private	Private
Number of beds	400 beds	263 beds	517 beds
Number of employees	1055 people	374 people	1326 people
Number of departments	26 departments	21 departments (8 wards)	45 departments (16 wards)
Special beds for infectious diseases	No	No	No
Emergency department	Yes	No	Yes

**Table 2 ijerph-18-10226-t002:** Analytical framework.

Domain	Check Point
Operational management	Establishment of COVID-19 task force and clarification of the chain of command. Establishment of cooperative system with PHCs. Establishment/strengthening of regional medical network (replenishment of human resources, cooperative system for accepting infected patients, etc.). Appropriate allocation of human resources. Transfer of infected patients/staff to hospitals/quarantine facilities.
2. Identification of infection status	Identification of outbreaks of infectious diseases (identifying suspected areas of infection). Identification of transmission routes by contact tracing those in close contact with infected persons, and monitoring health condition of all patients and staff. Quarantine and isolation arrangement for staff having close contact with infected persons. Conducting PCR testing. Close observation of the health condition of all staff members.
3. Infection control measures	Education on infection control to all staff. Minimalizing number of staff in contact with infected patients. Securing appropriate zoning and traffic flow. Conducting strict cohorting. Securing implementation of standard precautions and precautions by infection route (how to put on and take off personal protective equipment). Proper placement of dedicated items, securing of personal protective equipment, etc. Maintaining a clean and non-contaminated environment, appropriate medical waste management. Improvement of the work environment (measures to prevent cross-infection among staff).

PHC, public health center; PCR, polymerase chain reaction.

**Table 3 ijerph-18-10226-t003:** Timeline of events at Hospital A.

Date	Day	Event
21 March	−2	Two patients in Ward A on the fifth floor with fever were tested for COVID-19 using PCR. New admissions to that ward with the two suspected patients were suspended.
23 March	0	The two above-mentioned inpatients were found to be positive by PCR testing (Index case).
24 March	1	One patient and one nurse with symptoms were found to have positive PCR tests. Quarantine of all healthcare workers (HCWs) in the ward was imposed. Outpatient services and new admissions in the ward with COVID-19 patients were suspended.
25 March	2	Nine patients were found to be COVID-19 positive by PCR test (on which floor is unknown). New admissions in all wards was suspended.
26 March	3	PCR testing was performed on all patients in all wards.
27 March	4	PCR testing was conducted on all staff. COVID-19 positive patients were isolated, staff in close contact with the COVID-19 suspected patients were self-quarantined at home New staff were assigned to all wards.
28 March	5	Floors 5–8 have been designated as contaminated areas. Wearing PPE was made mandatory for all staff entering floors 5–8.
30 March	7	The MOHLW cluster team was dispatched. Field survey and support was initiated. Rearrangement of cohorting and zoning by wards was initiated.
1 April	9	Frequent reassignments of nurses were made due to frequent isolation of nurses.
9 April	17	The rearrangement of cohorting and zoning was completed.
15 April	23	The interim report by the MOHLW cluster team was published.
9 May	47	The total number of infected cases at that time was 192, including 109 inpatients and 83 staff.
26 May	64	Follow-up outpatient service has been resumed.
6 June	75	New admissions to the internal medicine and surgical wards were resumed.
8 June	77	The containment was declared on the hospital website. Hospital A resumed accepting new inpatients by separating the PCR-positive and post-healing patients,
19 June	88	Emergency department (except at night) was resumed.

PCR, polymerase chain reaction; PPE, personal protective equipment; MOHLW, Ministry of Health, Labour and Welfare.

**Table 4 ijerph-18-10226-t004:** Timeline of events at Hospital B.

Date	Day	Event
6 November	0	One inpatient and one staff member on the sixth floor were found to be PCR- positive (Index case). The PHC was contacted. A taskforce team was set up in the hospital according to the PHC’s instruction. The PCR test of 21 patients and 4 staff members on floor 6 were found to be positive at midnight.
7 November	1	Under the guidance of the PHC, the taskforce team was enhanced with four outside experts and five hospital staff members.
8 November	2	The positive patients were transferred to the designated hospital.
9 November	3	Outpatient and routine health-checkup examination services were suspended.
11 November	5	A positive case occurred on floor 7.
15 November	9	A positive case occurred on floor 5. Wearing PPE was mandated on floors 5–7. The physicians’ ward rounds have been reduced by the PHC instruction. Staff zoning in backyard and fixing of assigned inpatients were introduced.
16 November	10	Since mid-November, the number of staff in charge of wards has been reduced by a third, and the shortage of HCWs has become more serious. The shortage of supplies also became more pronounced.
17 November	11	PCR testing of ward staff on floors 5–7 was conducted. Requests for outside support were made due to Insufficient supplies.
19 November	13	During Days 10 to 13, five chief ward nurses tested positive by PCR and were isolated at home, so the physician in charge resumed ward rounds. It became more difficult to find transfer hospitals for COVID-19 patients. The beds at the health check-up service department were provided for infected staff.
20 November	14	In Days 14 to 18, the first expert team was dispatched. Monitoring and evaluation of the hospital’s NI outbreak status and IPC measures were carried out. Zoning and cohorting were strengthened, individual infection control practices were strictly taught by on-the-job training and supervision.
21 November	15	Disinfection of infectious wards by UV irradiation was conducted from Day 15 to 17.
22 November	16	Large-scale NI outbreaks occurred at other two hospitals in Asahikawa City.
24 November	18	The infection has spread to all wards. The PHC ordered the hospital to switch from PCR to antigen testing due to the shortage in the city’s PCR testing capacity.
25 November	19	The hospital representative met with the mayor of Asahikawa City to request assistance, but was declined. On the same day, the representative sent a written request to the Governor of Hokkaido.
27 November	21	The second expert team was dispatched. The expert team conducted another evaluation of current IPC measures. They established a centralized communication mechanism with all departments in the hospital and held regular meetings twice a day. The outsourcing service of cleaning was suspended.
29 November	23	Relief supplies from outside began to arrive.
30 November	24	A demonstration of putting on and removing off PPE and intensive practical training on IPC precautions were given by a professor of HSU.
1 December	25	The interim report on the hospital outbreak was published on the hospital website. With DMAT arrangements, five designated hospitals in the city agreed to receive infected patients.
3 December	27	A meeting of the directors of the five main hospitals in the city was held, and it was agreed that 10 patients would be transferred to these hospitals.
6 December	30	As the transfer of positive patients still did not materialize, the hospital decided to keep positive patients and provide care in the hospital. The fifth and sixth floors were designated as a red zone and the fourth and seventh floors into a green zone, which made it possible to do triage within the hospital.
8 December	32	The Self-Defense Forces nurse (until Day 45) and outside nurses (until Day 66) were dispatched.
9 December	33	The hospital outbreak seemed to be on the path to containment.
19 December	43	From Day 30 to 43, the detailed symptoms of all patients by wards were identified and a list of patients by bed/ward was made. UV irradiation, cleaning, and disinfection were carried out to thoroughly maintain the virus-free environment.
21 December	45	The DMAT support ended.
23 December	47	The PHC supports ended.
3 January	58	The last confirmed case of infection occurred.
18 January	73	Under the guidance of the PHC, part of outpatient services were resumed
30 January	85	Hospital B declared containment on the website. Outpatient services (except rehabilitation), new inpatients, and health-checkup services were resumed.

PHC, public health center; PCR, polymerase chain reaction; HCWs, health care workers; NI, nosocomial infection; UV, ultraviolet; PPE, personal protection equipment; IPC, infection prevention control; HSU, Health Service University of Hokkaido; DMAT, disaster medical assistance team.

**Table 5 ijerph-18-10226-t005:** Timeline of events at Hospital C.

Date	Day	Event
27 November	0	One staff member was found to be PCR-positive (Index case). New admissions, transfers in and out the relevant wards were suspended.
18 December	21	Six staff members and five inpatients tested positive by PCR. A rapid spread of infection among inpatients was observed. New admissions to the ward were suspended, and night and holiday treatment was suspended.
22 December	25	Days 24–25, 13 staff and 14 inpatients tested positive by PCR. Admitting new inpatients in all wards and accepting new outpatients was suspended. Re-examination of outpatients was sustained.
24 December	27	Follow-up of discharged patients has been initiated.
28 December	31	On Days 26–31, 46 staff and 48 inpatients tested positive by PCR.
3 January	37	On Days 32–37, 53 staff and 14 inpatients tested positive by PCR.
4 January	38	The hospital requested support from the prefecture.
6 January	40	A joint taskforce team was established. Zoning and other measures were initiated, and staff were instructed to wear PPE.
9 January	43	The hospital required outsourced staff to wear protective gear.
15 January	49	An interim report by the taskforce team was released.
16 January	50	Reorganized zoning and instruction on IPC measures have been initiated.
17 January	51	It was agreed to dispatch about 20 support staff from other hospitals in collaboration group. Guidance on infection prevention was started for staff returning to work from isolation.
29 January	63	The last new case of infection (one staff member) occurred.
1 February	66	Outpatient laboratory service was resumed.
26 February	91	The convergence was declared on the hospital website.
1 March	94	Normal medical care resumed.

PCR, polymerase chain reaction; PPE, personal protection equipment; IPC, infection prevention control.

**Table 6 ijerph-18-10226-t006:** Summary of remarkable risk factors in the spread of NI and factors in its prevention in the three hospitals.

Hospital	Operational Management			Identification of Infection Status		Infection Control Measures				
	External Supports	HCW Resource Allocation	Inpatient and Outpatient Control	Epidemiological Analysis	PCR Test	Zoning and Cohorting	Staff IPC Practices	Staff-to-Staff Transmission	Environmental Cleaning and Disinfection	Transferring Infected Patients
A	Immediate external assistance improved overall situation		Immediate cessation of both inpatient and outpatient services	(a) Delayed recognition of COVID-19 positive case	Long wait for results	External expertise was needed to implement appropriate zoning and cohorting	Intensive training on handwashing, selection of appropriate PPE how to properly wear and remove PPE, how to manage medical disposure for staff who work in wards	Low recognition of staff-to-staff transmission risk	When changing zones, to ensure the green zones disinfecting environment by sanitization	-
B	Immediate, repeated and intensive external assistance improved overall situation	Requests for municipal and prefectural assistance denied	Immediate cessation of outpatient services	(a) Late recognition of NI; (b) no identifying virus route; (c) not preparing an inpatients monitoring list	Limited testing capability	External expertise was needed to implement appropriate zoning and cohorting	Repeated and intensive training on full IPC measures and supervision by external experts	Low recognition of staff-to-staff transmission risk and high prevalence	(a) Instruction on cleaning and disinfecting wards and toilets, (b) frequent UV exposure	Very limited capacity for transferring due to high CAI outbreak
C	Delays in contacting the PHC and receiving external support caused the large-scale outbreak	Poorly allocated due to large number of infected HCWs and little outside assistance	Gradual and partial cessation of inpatient, outpatient and emergency room services	(a) Late recognition of NI, b) no identifiable virus route	-	External expertise was needed to implement appropriate zoning and cohorting	Training on comprehensive IPC for medical staff and co-medical staff, including how to properly wear and remove PPE	Low recognition of staff-to-staff transmission risk and high prevalence	(a) Instruction on collection and disposal of infected waste	No transfers due to high CAI outbreak

HCWs, health care workers, PCR, polymerase chain reaction; IPC, infection prevention control; PPE, personal protection equipment; UV, ultraviolet; CAI, community-acquired infection; PHC, public health center; NI, nosocomial infection.

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
