# Peer review of "Overcoming Current and Preventing Future Nosocomial Outbreaks during the COVID-19 Pandemic: Lessons Learned at Three Hospitals in Japan"

_ijerph, 2021, doi:10.3390/ijerph181910226_

Round 1
Reviewer 1 Report
Table 3. I got a little bit confused. Who are the index patients indeed? The two patients with fever in March 21? Or one patient and the nurse with symptoms in March 24? Or, did the nurse have fever in March 21? Please clarify.
Table 3. March 28, Day 5. I am a bit surprised that the staff (at least) were not wearing appropriate PPE. What were the biosafety recommendations given some experience of the COVID-19 outbreaks in China and the beginning of the outbreaks in Japan since January 24, 2020?
Page 6, lines 189-190. What were the biosafety recommendations for patients and staff moving inside the hospitals? What were the recommendations for those with suspected SARS-CoV-2 infection? I think that would help to clarify the spread. (I think the question could be applied to all hospitals)
Reviewer 2 Report
Please make sure there is consistency in how you refer to the days. Sometimes it is not clear if the dash is to indicate a negative or to serve as a connection. Consider if it may be helpful to use the dash only to indicate negative days from the index case (e.g., Day -4 = 4 days prior to index case; Day 12 = 12 days after index case).
The English style and grammar can be improved upon, particularly in the figure captions and tables. Table 6 in particular needs editing.
The timeline and figures are very helpful and effective. However, I am somewhat confused with the timeline, given that you reported that the PCR results were not available until much later in most cases. So, for example, how is it that on Day 0 for Hospital C that new admissions and transfers were suspended? Perhaps Day 0 is not when the index case occurred but rather when the PCR test results returned? Also, why is the index case for Hospital A not on Day 0? You may need to adjust the timeline or provide more explanatory notes in order to ensure there are not conflicts.
Consider if the discussion may benefit from being organized by the three major categories you had previously defined.
Consider including a table summarizing helpful strategies that were used by these three hospitals or other hospitals you evaluated during your review. You could perhaps include pitfalls observed with corresponding solutions.
I recognize that the nature of this manuscript is to provide qualitative data regarding infection prevention management within these hospitals. However, consider if some type of scoring system may be able to be developed for the three categories you described. If not a numerical score, could use a relative scale, for example, excellent, adequate, or poor response.
